# Localization of Free and Bound Metal Species through X-Ray Synchrotron Fluorescence Microscopy in the Rodent Brain and Their Relation to Behavior

**DOI:** 10.3390/brainsci9040074

**Published:** 2019-03-28

**Authors:** Caroline L. C. Neely, Stephen L. P. Lippi, Antonio Lanzirotti, Jane M. Flinn

**Affiliations:** 1Department of Psychology, George Mason University, 4400 University Drive, Fairfax, VA 22030, USA; jflinn@gmu.edu; 2Department of Psychology & Sociology, Angelo State University, 2601 W. Avenue N, ASU Station #10907, San Angelo, TX 76909, USA; stephen.lippi@angelo.edu; 3Center for Advanced Radiation Sources, University of Chicago, 9700 South Cass Avenue, Argonne, IL 60439, USA; lanzirotti@uchicago.edu

**Keywords:** zinc, copper, iron, long-term potentiation, X-ray fluorescence

## Abstract

Biometals in the brain, such as zinc, copper, and iron, are often discussed in cases of neurological disorders; however, these metals also have important regulatory functions and mediate cell signaling and plasticity. With the use of synchrotron X-ray fluorescence, our lab localized total, both bound and free, levels of zinc, copper, and iron in a cross section of one hemisphere of a rat brain, which also showed differing metal distributions in different regions within the hippocampus, the site in the brain known to be crucial for certain types of memory. This review discusses the several roles of these metals in brain regions with an emphasis on hippocampal cell signaling, based on spatial mapping obtained from X-ray fluorescence microscopy. We also discuss the localization of these metals and emphasize different cell types and receptors in regions with metal accumulation, as well as the potential relationship between this physiology and behavior.

## 1. Introduction

Metal homeostasis has gained attention within the past few decades due its involvement in numerous biochemical interactions and diseases, such as Wilson’s disease, Menke’s disease, Alzheimer’s disease, depression, and anemia [1,2,3,4]. These diseases and conditions are not limited to one particular type of neurodegenerative process or cognitive or behavioral deficit; rather, altered biometal concentrations in the brain can induce impairment across a broad spectrum of behaviors such as learning and memory. Disruptions in these biometals during embryonic development and adulthood can result in oxidative damage as well as disruption to signaling pathways, specifically those involved with long-term potentiation (LTP) in the hippocampus [5,6]. Although biometals are often described as “trace” in the brain, there are substantial concentrations in different regions of the hippocampus, which in part may explain selective effects of these metals in learning and memory. 

In this review, we focus on the different regions of the hippocampus and their associated metals as demonstrated by X-ray fluorescence microscopy. We start with a description of hippocampal circuitry involved in LTP and some of the cell types that underlie the circuit’s role in physiology and behavior. We briefly explain our use of X-ray fluorescence microscopy to obtain images of zinc (Zn), copper (Cu), and iron (Fe) within coronal rat hippocampal sections. We then describe the localization of Zn, Cu, and Fe with particular emphasis on our obtained images of coronal rat brain hippocampus. In this context, we initially review studies that demonstrate the profound effects each biometal has on the functionality of the trisynaptic circuit that underlies learning and memory. We provide a review of the trisynaptic circuit and receptor types, the influence of these biometals on LTP, and behavioral correlates. The understanding of physiological conditions of these biometals may shed light on pathological conditions, as dyshomeostasis in these metals can result in alterations in hippocampal-dependent behaviors and electrophysiological properties [7] as seen in several neurological and neuropsychiatric disorders. 

## 2. The Hippocampal Trisynaptic Circuit and Long-Term Potentiation (LTP)

Trace biometals are found in comparatively small quantities but have important functional roles across different cell types in the brain. One critical area where these biometals exert important signaling and plasticity effects is in the hippocampus, a limbic structure that is now known to be responsible for the formation of certain types of memory. This trisynaptic circuit was first illustrated and described in the 20th century by neuroanatomist Santiago Ramón y Cajal using the Golgi-Cox method [8]. This relay circuit includes connections which are involved in information processing and explicit memory of mammals [9]. The overall pathway begins in the entorhinal cortex and terminates in the subiculum [10]. Initially, axons of pyramidal cells in the entorhinal cortex form the perforant pathway to granule cells of the dentate gyrus. Granule cell axons, the “mossy fibers”, synapse with pyramidal cells in Cornu Ammonis subfield CA3; in turn, axons of CA3 pyramidal cells project to CA1 pyramidal cells via Schaffer collaterals. Some projections from CA3 also leave via the fornix and loop around the thalamus where they connect to the hypothalamus. Finally, axons from CA1 pyramidal cells synapse with stellate and pyramidal cells in the entorhinal cortex and subiculum. Figure 1 illustrates the organization of the hippocampus, as originally drawn and adapted from Ramón y Cajal (1911) [8]. 

Much of what is known today regarding the direction and flow of impulses throughout the circuit were correctly inferred by Ramón y Cajal himself [12]; however, it was the experiments in rabbit hippocampus conducted by Lømo, Andersen, and Bliss in the late 1960s and early 1970s that led to the characterization of LTP and its role in learning and memory [13,14]. Lømo applied stimulation to the perforant pathway and observed excitatory postsynaptic potential (EPSP) responses in the dentate gyrus. When a high-frequency train of stimulation was applied to the presynaptic side, a long-lasting increase in excitatory postsynaptic amplitude and an increase in number of spikes were observed during each train. In addition, the first spike amplitude remained potentiated and stronger EPSPs could be generated from the same single stimulation of the presynaptic neuron population [15]. Later work by Bliss confirmed that this high-frequency stimulation could enhance synaptic transmission of the rabbit hippocampus for days and weeks, suggesting that alternative mechanisms, other than cell membrane excitability, were capable of driving long-lasting changes in neurons. Lømo coined the term “long-term potentiation” specifically in the perforant pathway in the hippocampus to reflect that “cells that fire together, wire together” [16,17].

Long-term potentiation is now generally studied in the CA1 region of the hippocampus where it is thought to depend on the various receptors that bind glutamate. Glutamate has several receptor types, three of which are ionotropic: the α-amino-3-hydroxy-5-methyl-4-isoxazolepropionic acid (AMPA) receptor, the kainate receptor, and the N-methyl-D-aspartate (NMDA) receptor. Glutamate also binds to its metabotropic receptor family, mGluRs, which mediate intracellular cascades via interactions with G-protein coupled receptors. Glutamate partially activates the NMDA receptor (NMDAR), a ligand-gated and voltage-sensitive receptor paired to a channel permeable to sodium and calcium (Ca^2+^) ions. NMDARs require both glutamate binding and slight depolarization to remove the magnesium ion plug from its pore, allowing sodium and Ca^2+^ entry [18,19]. Calcium entry subsequently activates several protein kinases that are essential for the maintenance of LTP. They phosphorylate existing sodium-permeable AMPA receptors (AMPARs) and increase conductance of these AMPA ion channels. In addition, Ca^2+^ entry via NMDA activation leads to the activation of Ca^2+^/calmodulin-dependent protein kinase II (CaMKII) that promotes the embedding of GluR1-containing AMPARs into the neuronal membrane near the activated synapse [20]. The embedding and reversible phosphorylation of AMPARs mediate the maintenance and sustainability of LTP in the CA3-CA1 Schaffer collaterals [21]. Indeed, several experiments have demonstrated a causal link between NMDARs, AMPA insertion, and learning and memory, albeit these are not the only mechanisms. Selective deletions of the *NMDAR1* gene from CA1 neurons, application of NMDA antagonist AP5, and deletions and mutations in *CaMKII* genes impair NMDAR currents, LTP, and spatial memory in mice and rats [22,23,24,25]. Whitlock and colleagues’ (2006) study [26] also elucidated the causal relationship between LTP and learning. Thirty minutes after a single training session of inhibitory avoidance in rats, AMPAR delivery in CA1- and NMDAR-dependent phosphorylation of serine 381 in AMPA receptors were increased, both of which are considered markers of LTP. These were accompanied by increases in field EPSP slopes compared to those of naive rats and yoked-control rats. Whitlock and colleagues suggested that these hippocampal changes, i.e., CA1 LTP, were induced by inhibitory avoidance training in rats. One should warrant caution, however, as the presence and increase of AMPARs cannot solely explain the phenomenon that is LTP [27,28]. Complex AMPA protein–protein interactions and activity-dependent responses in different regions of the circuit, rather than solely AMPA insertion itself, may explain behavioral changes in learning and memory.

NMDA-dependent LTP was originally studied in the perforant pathway in rabbit hippocampus; however, LTP between Schaffer collateral axons that leave the CA3 region and synapse onto CA1 pyramidal cells is NMDA-dependent [29] and remains the most largely studied as the prototypical model of LTP [30]. Other non-NMDA mediated mechanisms that underlie LTP, i.e., NMDA-independent LTP, have been discovered to occur primarily at mossy fiber projections. The mossy fiber projections to the CA3 region are Ca^2+^- and NMDA-independent [19,31,32,33,34,35] and may involve alternative mechanisms such as metabotropic glutamatergic receptors, kainate receptors, or opioid receptors [36,37]. It is important to note, however, that NMDARs are still present at mossy fiber/CA3 synapses, and although their functional role in the prototypical model of LTP is unclear, it is believed that these NMDARs may partially mediate postsynaptic depolarizations involved in LTP, but at a lower intensity compared to AMPARs [38,39]. Alternatively, a mechanistically different form of LTP at mossy fibers has been proposed: LTP at mossy fibers is shown to increase NMDAR expression through a Ca^2+^-dependent, protein kinase C-mediated mechanism [40]. Despite the uncertainty of how these receptors are involved in LTP, CA3 NMDARs remain crucial to learning and memory, and disruptions in gene encoding and expression can lead to impairments in one-trial learning [41] and paired associative learning [42]. Lastly, LTP can also occur in other brain regions outside the hippocampus such as the amygdala [43] and in many areas of neocortex [44,45,46]. 

Biometals, such Zn, Cu, and Fe, are known to interact with NMDARs, AMPARs, and other hippocampal cell types and are known to be involved with cofactor activity, modulation of synaptic receptor activity, neurotransmitter synthesis, and neural plasticity associated with LTP. The remainder of this review explains the localization of Zn, Cu, and Fe in the hippocampus, as its different regions are seen to have differing metal concentrations, and to provide a summary of research that focuses on each metal as it pertains to hippocampal cell types and normal and abnormal behavior. The metal distributions in other brain regions outside of the hippocampus are briefly discussed. 

## 3. Metal Localization and Their Roles in Learning and Memory

### 3.1. Imaging Methods to Infer Localization 

We used Synchrotron X-ray fluorescence microscopy (Beamline X-26A, National Synchrotron Light Source, Brookhaven National Laboratory) to spatially localize Zn, Cu, and Fe at concentrations below 100 parts per million (ppm) within the trisynaptic circuit in rat hippocampus (Figure 2). This method is advantageous because it allows for the visualization of total, i.e., both free and bound, metal ions (with the latter invisible to most histological stains) at a spatial resolution of 5 μm or smaller without destroying the tissue. This technique generally does not destroy tissue, a major disadvantage of other methods such as laser ablation inductively-coupled mass spectrometry or particle induced X-ray emission. In addition, localization allows for researchers to infer the role these metals play in brain structures involved in certain cognitive processes. For the purpose of this review and transparency regarding how we obtained our images, we briefly describe the imaging process as follows. Samples were obtained from a 4-month-old male Sprague–Dawley rat, were sliced into 40 μm sections, and mounted onto metal-free, Suprasil quartz slides (Heraeus). X-ray microfluorescence images were collected by scanning the section through a focused X-ray beam and measuring the emitted X-ray fluorescence. The incident beam was focused to a spot size of 10 μm using a pair of 100 mm long grazing incidence mirrors in a Kirkpatrick-Baez geometry. Fluorescence counts were measured using an energy dispersive, 9 element Germanium solid-state detector using dwell times of 5 s per pixel and a step size of 40 μm. The X-ray fluorescence maps demonstrate Zn, Cu, and Fe Kα emission lines at each pixel normalized by incident beam intensity measured in an ion chamber upstream of the focusing optics.

Metal distribution can be verified via basic histological stains, one of which is thionin, a common and simple stain that can be performed in fresh-frozen tissue (as demonstrated in Figure 3). Structure location should be verified using a rat brain atlas [47] and the Allen Institute Mouse Brain Atlas [48].

### 3.2. Properities and Localization of Zinc

Zinc is the second most abundant biometal in the body and brain and is involved in more than 300 enzymatic reactions [6,49]. Zinc is obtained through the diet and is absorbed and transported through the intestine by an unknown intracellular pathway [50]. Approximately 2%–3% of total Zn resides in tissues such as the brain, which it relatively quickly accumulates near the fourth ventricle and brainstem regions and then spreads to more rostral areas, such as the hippocampus [51,52]. At the cellular level, three pools of Zn have been identified: (1) the vesicular pool; (2) the membrane-bound, or protein-metal complex pool; and (3) the ionic or “free” Zn pool. The vesicular pool refers to Zn ions that are loaded into round, clear vesicles and are often released with glutamate at “gluzinergic” synapses [53]. However, Zn also colocalizes with GABA and glycine [54]. The amount of free Zn present during an action potential has not been accurately measured due to the limitations of current imaging techniques, but it is hypothesized that free Zn levels in the synaptic cleft can rise to 1–100 µM, with higher amounts up to 300 µM from hippocampal neurons under normal activity [55,56,57]. Clearance mechanisms remain unknown, with several theories suggesting that Zn diffuses away from the synapse, or binds with high affinity to postsynaptic targets, forming a Zn veneer until other mechanisms clear it away [58]. 

Zinc primarily modulates postsynaptic ionotropic and metabotropic receptors through Zn-specific allosteric binding sites and acts as a second messenger in the cytosol [52,54]. Stimulation of specific neurons, sometimes referred to as gluzinergic neurons, leads to the co-release of glutamate and Zn, modulating several receptors such as NMDA, AMPA, and GABA_A_, and those linked to Ca^2+^ channels [59,60,61]. Low stimulation and low levels of Zn tend to antagonize NMDA channels by binding to the NR2A subunit and pore-lining residues within the channel and protect against NMDA toxicity; in contrast, higher levels of stimulation promote Zn/glutamate corelease, causing glutamate to activate extrasynaptic NMDARs that are not reached by synaptically-released Zn [62,63]. In addition, evidence has shown that Zn modulates GABAergic signaling by reducing amplitude, decreasing onset rate, and accelerating the decay rate of postsynaptic currents at GABA_A_ receptors [64,65]. These results show that Zn is heavily involved with the glutamatergic neuromodulation, such that simultaneous Zn/glutamate release can both promote and inhibit NMDA-receptor activity while potentiating non-NMDA synaptic activity [66,67]. Frederickson and colleagues (2000) [68] further review the role of Zn in neurotransmission, emphasizing Zn-containing neurons in forebrain structures including the amygdala, hippocampus, and cortex.

The presence of Zn in the hippocampus can be noted in the composite Figure 2 and in Figure 4. The strongest Zn signal is seen in the granule cell layer of the dentate gyrus and extends to the CA3 pyramidal cell layer. The molecular layer of the dentate gyrus is nearly devoid of Zn signal. There is some signal in the stratum radiatum but the signal dissipates in the CA1 region. The Zn localization overlaps prominently with regions that contain a high density of NMDARs, including parts of the CA3 and CA1 fields [10]. Zinc is packaged into vesicles by the transporter, ZnT3, and is stored and released with glutamate in gluzinergic neurons [69,70]. This vesicular Zn transporter is prominent in the terminals of mossy fibers and is also found in the cerebral cortex and other limbic structures [49], which may explain the strong signal emitted from the mossy fiber connections as well as the intermediate signal in overlying cortex. Our obtained image specifically localizing Zn to these cell layers is consistent with prior studies that have localized Zn using alternative methods. For example, Zinpyr-1, a common fluorophore used as a marker for free, unbound Zn, has been used to illustrate this similar band of staining along the mossy fiber pathway from the granule cells in the dentate gyrus as well as the Schaffer collaterals [71,72,73,74]. Outside of the hippocampus, notable staining can be seen at the end of the external capsule and nearby cortex which correspond with the amygdala and the piriform cortex, respectively. Previous studies have confirmed higher concentrations of histochemically reactive Zn concentrations in nearly all amygdaloid nuclei, although specific regions, such as the lateral amygdala and the amygdalo-piriform transition area [75,76] contain high concentrations of Zn.

#### Zinc, LTP Signaling, and Learning and Memory

It is possible that Zn exerts effects within the trisynaptic circuit by interacting with different receptors. Several compounds such as Clioquinol (CQ) have been used to sequester Zn and effectively negate any effects in order to understand Zn signaling in LTP [77,78]. Clioquinol administration into the dentate gyrus of rats has also been shown to disrupt LTP induction as well as LTP that had been maintained for 6 days. Co-injection of ZnCl_2_ with CQ remediated LTP, further demonstrating the involvement of Zn in LTP maintenance [78]. A recent study by Takeda and colleagues (2018) [79] demonstrated that age interacts with Zn regarding LTP maintenance. When given high (i.e., 1 μL 100 mM) K^+^ injections, aged Wistar rats exhibited increased Zn influx from the extracellular space into dentate granular cells, but young rats did not have this increase in Zn influx. There was no substantial influx in the CA1 or CA3 regions in either young or aged rats. The substantial increase in Zn influx in aged rats was accompanied by significant alterations in postsynaptic amplitudes indicative of LTP impairment. Co-administration of a Zn chelator, CaEDTA, into the dentate gyrus with a high K^+^ injection prevented Zn influx and rescued 6-day-maintained LTP. In addition, Zn influx and impairment were also rescued with co-injection of an AMPAR antagonist, CNQX. Thus, it is proposed that excess Zn influx via AMPARs may be responsible for alterations in LTP induction and maintenance [79]. 

Mossy fiber synapses also exhibit LTP but are NMDA- and Ca^2+^-independent; however, they also have high levels of Zn [80,81] and contain Zn-sensitive metabotropic receptors that activate MAPK and CaMKII pathways. These pathways include downstream effects such as intracellular release of Ca^2+^ and phosphorylation of extracellular-regulated kinase and Ca^2+^-calmodulin kinase [82]. Because Zn-sensitive CA3 NMDARs are still important for the learning and acquisition and consolidation of single-occurring events [41,83], it is proposed that Zn reshapes the NMDA response [80]. For instance, Li and colleagues (2001) [81] determined that relatively high levels of Zn (50–100 μM) induced long-lasting potentiation of EPSP response without high frequency stimulation. Zinc’s ability to induce long-lasting potentiation appears to be mediated by internalization into pre- and postsynaptic neurons. In the same study, application of antagonists of glutamate and voltage-gated Ca^2+^ channels in addition to a Zn ionophore permitted the internalization of Zn, leading to long-lasting potentiation. Removal of the Zn ionophore failed to increase internal Zn levels and subsequently failed to produce long-lasting potentiation [81]. In a similar study, Ceccom and colleagues (2014) [83] administered either or both NMDA-antagonizing AP5 and Zn-chelating CaEDTA into the CA3 of mice which subsequently impaired contextual fear learning in a 2-min, massed learning protocol. When conditioned and unconditioned stimuli were presented 2 h apart followed by drug administration, AP5 alone disrupted contextual fear learning whereas CaEDTA did not. However, joint AP5 and CaEDTA administration impaired contextual freezing compared to ZnEDTA control mice. 

Approximately 45% of NMDA-dependent Schaffer collateral boutons are Zn-positive [74] although Zn can originate from other sources other than synaptic vesicles [84]. Takeda et al. [84] utilized membrane-permeable and -impermeable Zn indicators to show that the Schaffer collateral-containing stratum radiatum of CA1 was poorly stained. In contrast, the dentate hilus and stratum lucidum, the general locations of mossy fibers of dentate granule cells, were strongly stained. When these regions underwent tetanic stimulation, fluorescence signal associated with both intracellular and extracellular Zn in the stratum radiatum (where Schaffer collateral synapses are present) was increased. As expected, tetanic stimulation of the Schaffer collaterals increased Ca^2+^ levels, but these levels were significantly attenuated by Zn application. Perfusion of CA1 using ZnCl_2_ in artificial cerebrospinal fluid also decreased glutamate concentrations as well, suggesting that Zn released from the Schaffer collaterals suppressed presynaptic and postsynaptic Ca^2+^ signaling followed by the suppression of glutamate release [84]. In conjunction to previous research, this study provides evidence that Zn serves as an inhibitor of NMDARs and glutamatergic signaling, albeit dependent on its concentration. Zn may also exert its effects via alternative receptors such as the extracellular P2X receptor, which is expressed in both epithelial cells and neurons [85,86,87] and has a role in the induction of LTP [88]. The role of the P2X receptor influences hippocampal circuitry through its Zn binding and subsequent Ca^2+^ influx in the presence of ATP and Zn: low concentrations of either Zn or ATP at the P2X receptor are shown to induce LTP within hippocampal CA1 slices, whereas high concentrations inhibit it. 

The broad range of enzymatic and synaptic activity in different brain regions signifies potential effects of Zn on a variety of behaviors. Several of these behaviors, such as fear conditioning and extinction, novel object exploration, and working memory, are hippocampal-dependent and have been shown to be sensitive to alterations in Zn concentrations and intake [27,77,78]. Takeda et al. (2006) [84] measured extracellular Zn and glutamate levels in the ventral hippocampus of rats exposed to novel environments for 50 min once a day for 8 days. Levels of glutamate were increased on days 1 and 2 of exploration, but by day 8, extracellular Zn levels had returned to basal levels. When extracellular hippocampal Zn was removed by CaEDTA, exploratory locomotion decreased, showing that presence of extracellular Zn impacted novel environmental exploration. Adlard et al. (2010) [89] demonstrated the importance of the Zn transporter, ZnT3, by using ZnT3 knockout (KO) mice. The mice showed impairments in the Morris water maze (MWM) task at six months and significant decreases in SNAP-25, PSD-95, and AMPAR densities, indicating poor synaptic connectivity. These behavioral deficits in ZnT3 KO mice may be due to a total lack of Zn in the dentate gyrus [90], further emphasizing the crucial role this transporter plays in vesicular packaging and subsequent downstream effects. Impairments in learning and memory due to Zn deficiency (and supplementation, as to be discussed) may be partially explained by interactions of Zn and neurogenesis within the dentate gyrus [91]. For example, CD-1 mice given Zn-deficient diets (0.85 ppm) had reduced TSQ fluorescence intensity in the CA1, CA3, and dentate gyrus, indicative of reduced hippocampal Zn levels. More importantly, this Zn reduction was accompanied by reductions in BrdU staining (a marker of cell proliferation) and immature neurons in the dentate gyrus [92]. The number of TUNEL-positive cells and the expression of many apoptotic-related proteins were increased as well, indicating that the severe reduction in hippocampal Zn availability negatively impacted plasticity [92]. Research by Suh et al. (2009) [93] yielded similar consequences on neurogenesis: Zn deficiency reduced both BrdU staining and DCX staining (a marker of developing/immature neurons) in the dentate gyrus of male Sprague–Dawley rats. Zinc depletion in the form of chelation has also been shown to reduce neurogenesis, even in cases of putative, compensatory neurogenesis, as seen following traumatic brain injury [94]. 

Likewise, Zn supplementation may also impair hippocampal-dependent behaviors. Supplementation with 10 ppm ZnCO_3_ in the drinking water of Sprague Dawley rats increased Zn in the hippocampus and impaired reference and working memory in the MWM; these learning and memory deficits were exacerbated in the older 9-month-rats condition [95]. The use of 10 ppm ZnCO_3_ has been shown to cause memory deficits in spatial memory tasks in wildtype mice and rats and Alzheimer’s disease transgenic mice in the MWM [96,97,98] and Barnes maze [99,100]. Interestingly, the use of ZnSO_4_ by various groups has had contradictory effects. Boroujeni and colleagues (2009) [101] used 10 ppm ZnSO_4_ through the drinking water and showed that Zn-enhanced rats spent more time in the target quadrant of the MWM compared to control mice when combined across training days. Additionally, Corona et al. (2010) [102] demonstrated that 3xTg-AD mice supplemented with 30 ppm ZnSO_4_ had shorter latencies to find a hidden platform in the MWM compared to transgenic mice on tap water, demonstrating an improvement in spatial memory.

Takeda (2012) [103] explored the effects of Zn in the hippocampus with particular attention to stress-induced attenuation of LTP by subjecting 6-week-old male Wistar rats to 30 s of tail suspension, which significantly increased serum corticosterone levels 1 h after tail suspension and attenuated CA1 LTP. Some rats received an injection of Zn-chelating CQ 2 h prior to tail suspension, which ameliorated the attenuation of CA1 LTP. These results show that Zn released with glutamate during times of stress may affect hippocampal LTP in a negative fashion, whereas Zn chelation may improve LTP.

### 3.3. Properities and Localization of Copper

Copper (Cu) is the third most abundant trace metal in the body, is obtained from the diet and absorbed from the gut. It promptly binds to albumin and is transported to the liver where it can be stored. Both ATP7A and ATP7B regulate Cu concentrations: ATP7A releases Cu into the portal vein of the liver in response to increased Cu levels, whereas ATP7B removes excess Cu via bile secretion [6]. Copper is mostly protein-bound and is tightly regulated through the nervous system; however, it passes through the blood–brain barrier via CTR1 and ATP7A transporters expressed in endothelial and choroid plexus cells [104,105,106]. Copper’s role in the nervous system is critical, particularly in the functioning of electron transfer associated with oxidative enzymes [107] and in oxidation-reduction reactions [105]. Copper is also included in a large number of cuproenzymes which depend on it for catalytic activity, energy production, immune functioning, free radical damage prevention, and neuronal transmission [105]. Levels of Cu in different brain regions are species-dependent and vary from individual to individual, but the dentate gyrus and CA1 of the hippocampus, the amygdala, the cerebellum, and parts of the diencephalon are generally cited as having the highest concentrations [105,108]. On a cellular level, Cu is concentrated at excitatory NMDARs and influences glutamatergic transmission. Copper remains trapped within glutamatergic vesicles and is released in a Ca^2+^-dependent manner, often exhibiting inhibitory effects on excitatory synapses [105]. Synaptic Cu has been shown to alter AMPA-specific signaling: short-term application of Cu resulted in a decrease in AMPA activity whereas a longer duration (3 h) led to enhanced AMPA activity in rat hippocampal neurons in vitro [109]. Copper influences glutamatergic and GABAergic signaling in brain regions associated with learning and memory, particularly spatial and associative learning, such as the hippocampus, the basolateral amygdala, associated intercalated cell masses, and the prefrontal cortex [65,95,108,110]. Activated NMDARs are essential for spatial-learning, and GABA receptor modulation in conjunction with NMDAR activity are important for extinction learning [65,110,111]. 

Composite Figure 5 demonstrates the localization of Cu surrounding the corpus callosum, the alveus just superior to the hippocampus, the linings of the third ventricle, and the choroid plexus. Increased signals surrounding the corpus callosum may be in part due to the role of Cu in maintenance of myelin integrity in the corpus callosum throughout adulthood [112,113]. Kuo et al. (2006) [106] localized Cu transporter CTR1 in the brain to the epithelial cells of the choroid plexus, which provides corroborating evidence for our obtained images (Figure 2 and Figure 5). Copper’s presence along the ventricular surface is also age-related in neurons and glia; thus, it is possible that our image obtained from a 4-month-old rat could yield different interpretations than those in younger or older animals. For example, Pushkar et al. (2013) [114] used X-ray fluorescence microscopy to show not only high levels of Cu in the subventricular zone (SVZ), a region important for neurogenesis, but also age-dependent increases in Cu content of SVZ astrocytes. Likewise, Fu et al. (2015) [115] found a relationship between age and Cu in the SVZ and metallothioneins (MTs) by examining Cu levels in Sprague–Dawley rats at age 3 weeks, 10 weeks, and 9 months. Nine-month-old rats had significantly higher levels of Cu in the SVZ, olfactory bulb, frontal cortex, and hippocampus and Cu transporters in the choroid plexus compared to both 3-week-old and 10-week-old rats. The choroid plexus and cerebrospinal fluid come into direct contact with the SVZ; thus, it is likely that the blood–brain barrier and cerebrospinal fluid systems that regulate Cu entry also explain the high concentrations of Cu within the SVZ [115]. However, the exact mechanisms of Cu interactions in neurogenesis and patterns of proliferation and migration remain poorly understood. Few studies have documented the role of Cu in neurogenesis, but interference with Cu regulation may disrupt normal migratory pathways. For example, Fu et al. (2015) [116] showed that Cu was highly concentrated in the SVZ of adult rats, and when rats were injected with 6 mg/kg manganese chloride, SVZ Cu levels were significantly reduced and DMT-1 transporter expression was increased. Manganese accumulation was accompanied by increases in GFAP-positive astrocytic stem cells and neuroblasts in both the SVZ and rostral migratory stream. Interference with Cu regulation via DMT-1/manganese exposure is one mechanism through which Cu is proposed to mediate neurogenesis; however, further exploration of other potential Cu transporters is warranted. Figure 5 provides a view of Cu accumulation faintly traced to the inner and outer portions of CA1 and CA2 of the hippocampus, with sparse signal arising from the dentate gyrus and no signal in the CA3 region.

#### 3.3.1. Copper, LTP Signaling, and Learning and Memory

Experiments that assess Cu’s effects on LTP and learning and memory often use dietary and/or drug treatments to induce Cu excess or deficiency in rodents and compare these data to those obtained from rodents on control diets with normal levels of biometals. For example, previous research has demonstrated that Cu alters electrophysiological properties of pyramidal cells of CA1 and alters the effectiveness of LTP in the hippocampus [117,118,119,120]. Rats given intraperitoneal injections (IP) of CuSO_4_ for 30 days failed to exhibit LTP after the MWM in contrast to rats that received saline [119]. Surprisingly, there were no significant differences in the MWM performance suggesting that, in this case, CA1 LTP was not a causal factor for learning and memory. In a follow-up experiment, rats were subjected to an initial training period and re-training period of the MWM, after which they were administered IP injections of saline or CuSO_4_. Although there was a single day during the re-training phase showing differences, there were no significant differences in latencies to find the hidden platform over all days. However, supplementation using different concentrations and delivery methods can impair LTP. CA1 LTP induction was impaired in rats 20 min after receiving a low (5mg/kg) IP dose or higher (10 or 15 mg/kg) IP doses of CuSO_4_ [120]. Leiva et al. (2009) [119] administered a low (1 mg/kg) IP dose of CuSO_4_ for 30 days, whereas Jand et al. (2017) [120] administered various concentrations of Cu and recorded 20 min post-injection. Although the administration of Cu was different between these two studies, both showed that low levels of Cu are capable of impacting LTP in the rat, specifically in CA1. These results show that alterations in hippocampal LTP may not always necessarily result in behavioral changes, i.e., in learning, although a large and growing body of evidence suggests otherwise [26,121,122].

#### 3.3.2. Interactions between Zinc and Copper

Copper deficiency can occur primarily by reduced dietary Cu intake: but severe Cu deficiencies can disrupt proper development, lead to premature death of neonates, or induce developmental effects that confound behavior that relies on movement to draw appropriate conclusions (i.e., freezing indicative of fear learning). As such, indirect or weaker Cu deficiencies are utilized to circumvent these developmental issues and include the use of Zn supplementation. Zinc actively competes with Cu in the gut and small intestine [123,124,125,126] and thus can be used to induce an indirect Cu deficiency. Sprague–Dawley rats that consumed water supplemented with 10 ppm ZnCO_3_ exhibited higher freezing rates during contextual retention, extinction, and cued extinction compared to animals on standard lab water at 4-months [98]. The same study noted similar MWM results compared to Flinn and colleagues’ work (2005) [95]: At 9 months, the rats raised on the Zn-supplemented water were tested in the MWM and exhibited longer latencies to reach the platform compared to controls, indicating impaired spatial memory. One group of Zn-supplemented rats were given Cu in their drinking water which remediated the spatial memory deficits in the MWM. These rats had similar results compared to the control rats, suggesting that results due to enhanced Zn may be in part due to a Cu deficiency. Similar results have been found with transgenic mice modelling Alzheimer’s disease where adding Cu to Zn-enhanced drinking water reduced the impairments due to Zn water [127]. These findings warrant caution, however, due in part to the intricate balance between Cu and Zn [6]. Positive or negative effects could be mediated by the chosen chemical composition of a Zn compound.

### 3.4. Properities and Localization of Iron

Iron (Fe) is the most abundant metal in the brain, and like Zn and Cu, it shares a variety of functions in biochemical activity, including, but not limited to, oxidative metabolism, myelination, and oxygen transport. Iron is obtained via the diet and is transported into the brain by a transferrin-mediated mechanism across the blood brain barrier [128]. Like other biometals within the body and brain, Fe concentrations are regulated by homeostatic mechanisms that prevent excessive Fe accumulation; in particular, as Fe concentrations increase, Fe binding-proteins are upregulated and transporters decrease [129]. Iron is tightly regulated by ferritin and transferrin, hemoglobin, heme enzymes and cytochromes, non-heme enzymes, and other iron-sulfur proteins [130]. Once inside the central nervous system, Fe is transported into neurons and supporting glial cells by transferrin- and ferritin-mediated processes, respectively [131,132], but the exact mechanisms of how transferrin and ferritin preferentially transfer Fe remain poorly understood (for an extensive review, see Lane et al., 2015 [132]). A free, or liable Fe pool exists within the mitochondria where it contributes to electron-transport systems and the synthesis of heme [133]; alternatively, Fe can be bound and stored away by ferritin. Although Fe is present throughout the brain, the basal ganglia and white matter structures consistently have the highest concentrations, and the overlying cortex and cerebellum have the lowest concentrations [134,135,136]. Differences in Fe concentrations across brain regions may be due to not only differences in Fe intake but also differing levels of ferritin expression as well. Mice given a low, 3 ppm Fe diet for one month had decreased Fe concentrations in the hippocampus and striatum compared to rats given a normal, 45 ppm diet [137]. Oligodendrocytes, the myelin-producing cells of the central nervous system, have been shown to stain for Fe particularly at the corpus callosum [138] and deficiencies in Fe have been seen to affect myelination in infants and rodents [139]. Iron deficiency was reported to be a leading cause of mild or moderate mental retardation despite its prevalence rate decreasing from the 1960s to the 1990s [140,141]; however, Fe deficiency still remains a risk factor for mental retardation in addition to many psychiatric disorders, including major depressive and bipolar disorders, autism, and attention deficit disorders [142].

The hallmark study by Hallgren and Sourander (1958) [134] has been consistently used as a standard for measuring Fe in the brain; however, new techniques such as X-ray fluorescence (as used for this review), inductively coupled plasma mass spectrometry, and quantitative susceptibility mapping, have been utilized [143]. The presence of Fe within the hippocampus is also inferred by previous research that has documented reductions in Fe staining in the CA1 field in mice lacking divalent metal ion transporter-1 (DMT-1) [144,145,146]. Specifically, when DMT-1 was removed from the hippocampus, there was lowered Fe uptake, reduced Fe in CA1 pyramidal cells, and altered CA1 dendritic morphology [144]. In our composite image (Figure 2), Fe appears limited within the CA1 field, with more prominent staining located in the striatum, subcortical structures, and cortex. Figure 6 shows clearer localization of Fe, with the strongest signals emitted from the pyramidal cells of the CA3 region, the granule cells of the dentate gyrus, and the basal ganglia. Some Fe is located in the pyramidal cells and the stratum lacunosum moleculare of the CA1 field. It is possible that the Fe signal in our image could originate from ferritin given that synchrotron X-ray fluorescence is sensitive to both free and bound Fe species.

Other work using X-ray fluorescence also showed Fe in the dentate gyrus and the CA1 field, providing corroborating evidence for our captured image [147]. NMDARs located in these regions may explain the presence of higher levels of Fe, as NMDAR activation leads to increased Fe entry into cells [148,149]. Iron also competes with Ca^2+^ entry through NMDARs and causes Fe accumulation, enhancing Fe release from lysosomes and alters the expression of Fe transporters [150]. Iron entry via NMDARs, despite competition with Ca^2+^ ions, is necessary for the regulation of glutamatergic excitability via PKC/Src/NR2A pathways [151]. Muñoz et al., 2011 [152] noted that NMDARs located in the CA1 field require Fe because Fe chelation disrupts LTP from occurring in this region. Muñoz et al. (2011) [152] utilized hippocampal cell cultures and electrophysiology to demonstrate the importance of Fe in neuronal CA1 LTP. The addition of desferrioxamine (DFO) and consequential removal of Fe led to a decrease in Ca^2+^ signaling via NMDAR activation. Application of DFO also caused altered phosphorylated ERK1/2 protein nuclear translocation as well as an inhibition in LTP induction in CA1 hippocampal neurons. Activation of ERK1/2 is a critical step in sustained LTP [152], which further demonstrates the importance of Fe within the hippocampus not only in its structural properties but also in its functional activities.

#### 3.4.1. Iron, LTP Signaling, and Learning and Memory

Iron deficiency during critical periods of development, including periods of hippocampal development, can lead to disruptions in neurogenesis, morphological development, axonal myelination, and catecholaminergic synthesis [153,154]. The concentrations of Fe and its related enzymes are the highest shortly after birth [155] and decrease upon maturation and closure of the blood–brain barrier and the start of myelination [156]. Like other metals, a dyshomeostasis of Fe, especially during early development, can impose long-lasting impacts to a variety of systems. Especially in early development, disruptions in Fe can decrease BDNF levels, which in turn, negatively impact cellular differentiation in the hippocampus [157]. These reductions in BDNF-mediated pathways may be compensated by alternative mechanisms such as GDNF, EGF, and NGF [157]. The CA1 region of the hippocampus contains Fe-dependent NMDARs, and disruption in Fe transport and Fe concentrations lead to problems in spatial memory acquisition [25], behavioral inhibition [158], T-maze performance, and fear conditioning [159]. Carlson and colleagues (2009) [144] noted that mice that lacked the DMT-1 Fe transporter in the CA1 field had significantly worse performance in the MWM task compared to mice with functional DMT-1. These results suggest that disruptions in CA1 translocation of Fe negatively impact spatial learning and memory. In another study, Felt et al. (2006) [160] administered a diet low in Fe to rat dams during their gestation period and later examined the offspring in a MWM task. The offspring exhibited impairments in spatial acquisition in the MWM, and these impairments failed to be remediated by a diet containing 40 ppm Fe that was later administered. A similar study done by Kwik-Uribe et al. (2000) [161] investigated the offspring of Swiss–Webster mouse dams given a diet low in Fe. Offspring that were continually fed the Fe-deficient diet failed to improve in the MWM compared to those given a Fe-repleting control diet. This Fe restriction also impacted the non-cognitive behavior of grip strength and startle responses: mice from dams that consumed a Fe-restricted diet displayed lower grip strength and startle responses compared to mice from dams that consumed a control diet. This could be due to decreased concentrations of Fe resulting in irreversible deficits in dopamine signaling within the basal ganglia [162,163]. Similar to the effects of low Fe and disruptions in Fe transport, Fe supplementation has also been shown to impair spatial memory, especially in mouse models of Alzheimer’s disease. Ten ppm iron supplementation through drinking water for approximately 11 months exacerbated spatial memory and reference memory of Tg2576 female mice, but had no deleterious effects in wild-type female mice [127]. 

The time frame during which Fe can impact development (and subsequent behavior) is not limited to gestation periods as previous reports suggest [160,161]. Rather, short-term Fe supplementation may also impact a wide range of cognitive and non-cognitive behaviors. For example, Maaroufi et al. (2009) [164] assessed Fe supplementation by intraperitoneally injecting adult Wistar rats with 3.0mg/kg FeSO_4_ for a period of five days and conducted MWM in addition to the open field test and elevated zero maze, which measure general locomotion and anxiety, respectively. Rats supplemented with Fe spent less time in the target quadrant in probe trials of the MWM, indicating a learning impairment. In addition, supplemented rats showed decreased ambulation in the open field task and increased time spent in closed arms in the elevated zero maze compared to controls, indicative of disrupted gross locomotion and increased anxiety. The hippocampus and basal ganglia of Fe-injected rats showed increases in brain Fe levels compared to control rats as well, establishing correlative evidence for Fe affecting these behavioral outcomes. Other work has also examined the correlations among the concentrations of Fe, hippocampal size, and memory performance. Rodrigue and colleagues (2013) [165] examined adult human subjects and found that with age, Fe levels in the hippocampus increased and hippocampal volume decreased. These findings together were related to memory performance of names and logical memory. 

#### 3.4.2. Interactions among Zinc, Copper, and Iron

Similar to the interpretations of Zn and Cu research, caution should be taken to avoid attributing behavioral changes to one metal alone. Excess Fe consumption in young weaned rats has been shown to cause systemic Cu deficiency anemia, low serum and tissue Cu levels, and decreased ceruloplasmin activity, and slowed body growth [166]. These systemic challenges were remediated by Cu supplementation in conjunction to the excess Fe, providing causal evidence of how depletion or supplementation of one metal can result in alterations of another. Additionally, these biometals share, and thus compete, for similar transport mechanisms and act on ionotropic channels. For example, the DMT-1 transporter has the capacity to transport Cu, Zn, and Fe. Espinoza et al. (2012) [167] used transfected human intestinal epithelium Caco-2 cells to inhibit DMT-1, and as a result, the uptake of Fe, Cu, and Zn decreased. This linear effect (as shown in one example of transport) is not as clear in regard to receptors and the consequences of their activation and inhibition. Synaptic effects on receptors are shown to be dependent on bind sites, receptor composition, and the concentrations of each metal, some of which have biphasic effects [168]. Therefore, experiments concerning any of these metals, especially those utilizing metal salts, should also consider potential interactions with other metals, both on a cellular and behavioral level. 

## 4. Conclusions

The trisynaptic circuit within the hippocampus is known to underlie learning and memory, and the brain regions that make up the circuit contain various concentrations of metals such as Zn, Cu, and Fe. In addition, the trisynaptic circuit has different receptors sensitive to these metals, and removal or supplementation of metals modulates these regions in both structural and functional ways. Zinc signal is strong in the ZnT3-containing mossy fiber projections between the dentate gyrus and CA3 field, a hippocampal region that exhibits LTP that is primarily considered NMDA-independent; however, as noted, a growing body of work suggests that a postsynaptic, NMDA-mediated LTP exhibited in mossy fiber connections exists, which may explain why some forms of learning (i.e., cue- and context-associative learning, one-trial learning) may be dependent on these present NMDARs, and thus, biometals. Other regions of the hippocampus, however, contain relatively higher amounts of Zn, although the dentate gyrus is cited as having the highest amounts. Zinc is involved with over 300 enzymatic processes, many of which pertain to LTP in the hippocampus. Likewise, Zn is also likely involved with synaptic connectivity in other regions outside of the hippocampus, such as the forebrain and neocortex. An increased Cu signal is seen around the ventricular spaces and the corpus callosum, although previous work has documented some amounts of Cu and Cu-dependent enzymes in the hippocampus. Localization to the corpus callosum is supported by evidence documenting Cu’s functional role in myelination. Localization to the subventricular zone also suggests that Cu has some role in neurogenesis. Lastly, Fe is the most abundant metal in the brain and is primarily found in white matter regions and the basal ganglia. Previous studies have localized some trace amounts of Fe to the hippocampus, and work utilizing Fe chelation demonstrates that Fe is required for hippocampal LTP and Ca^2+^ ion signaling. Iron deficiencies in early life can result in altered CA1 dendritic morphology as well as alterations in hippocampal-dependent behaviors. Overall, further exploration of Zn, Cu, and Fe and behaviors affected by alterations in these biometals will provide more understanding of neurodegenerative diseases, such as Alzheimer’s, Parkinson’s, and prion diseases.

## Figures and Tables

**Figure 1 brainsci-09-00074-f001:**
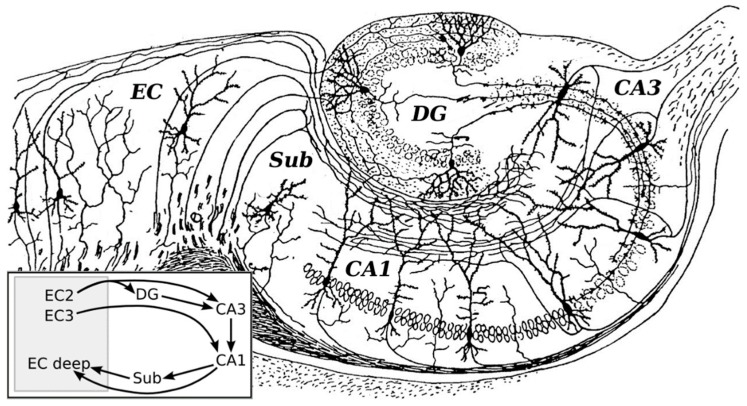
Organization of the hippocampal formation as originally drawn by Ramon y Cajal (1911) with a schematic representation of the layer organization. The trisynaptic circuit in the hippocampus is organised into three main regions: the dentate gyrus, the CA3, and CA1 fields. The perforant pathway connects the entorhinal cortex with the dentate gyrus and was the original site of study for long-term potentiation (LTP). Mossy fibers from the dentate gyrus project to the CA3 field. From CA3 fields, fibers may exit via the fornix (upper right) or continue to the CA1 via the Schaffer collateral pathway. The CA1 neurons then project onto other neurons in the entorhinal cortex and subiculum. EC = entorhinal cortex; CA = cornu ammonis; DG = dentate gyrus; Sub = subiculum. Modified image (rotated, relabelled, additional diagram in Gi) obtained from Wikimedia Commons Public Domain search [11].

**Figure 2 brainsci-09-00074-f002:**
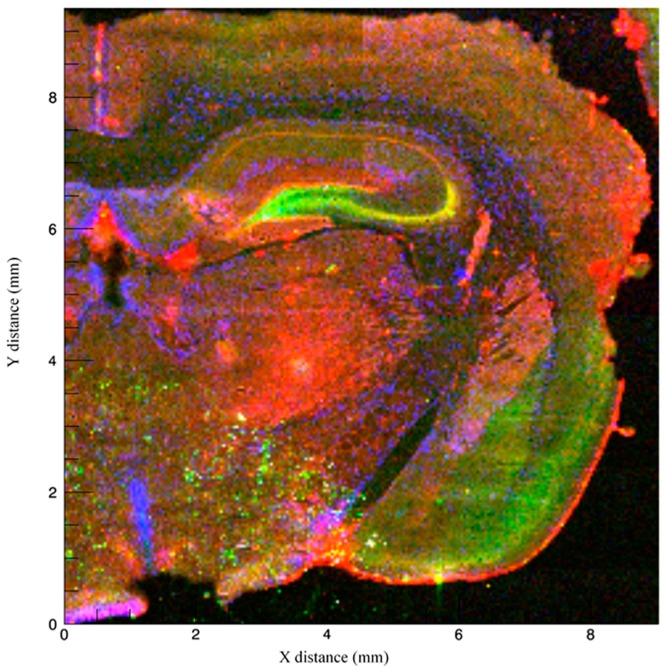
Coronal section of a 4-month-old Sprague–Dawley rat imaged using Synchrotron X-ray fluorescence, with fluorescence intensity of Zn (green), Cu (blue), and Fe (red) normalized to the incident X-ray intensity.

**Figure 3 brainsci-09-00074-f003:**
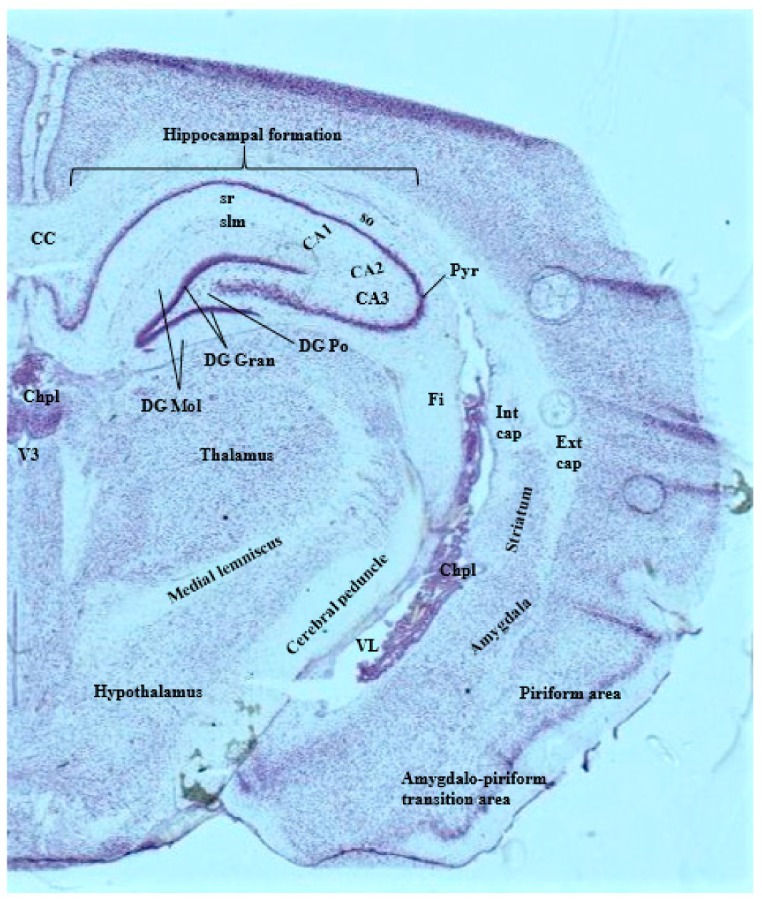
25 µm coronal section of a 4-month-old Sprague-Dawley rat brain stained with 0.5% thionin. Relevant brain structures and large landmark structures are labeled. CC = corpus callosum; CA = cornu ammonis fields; Chpl = choroid plexus; DG Gran, Mol, Po = Granule dentate gyrus layer, molecular dentate gyrus layer, polymorph dentate gyrus layer; Ext cap = external capsule; Fi = fimbria; Int cap = internal capsule; Pyr = pyramidal cell layer (CA1, CA2, CA3); slm = stratum lacunosum moleculare (CA1, CA2, CA3); so = stratum oriens (CA1, CA2, CA3); sr = stratum radiatum (CA1, CA2, CA3); VL = lateral ventricle; V3 = third ventricle.

**Figure 4 brainsci-09-00074-f004:**
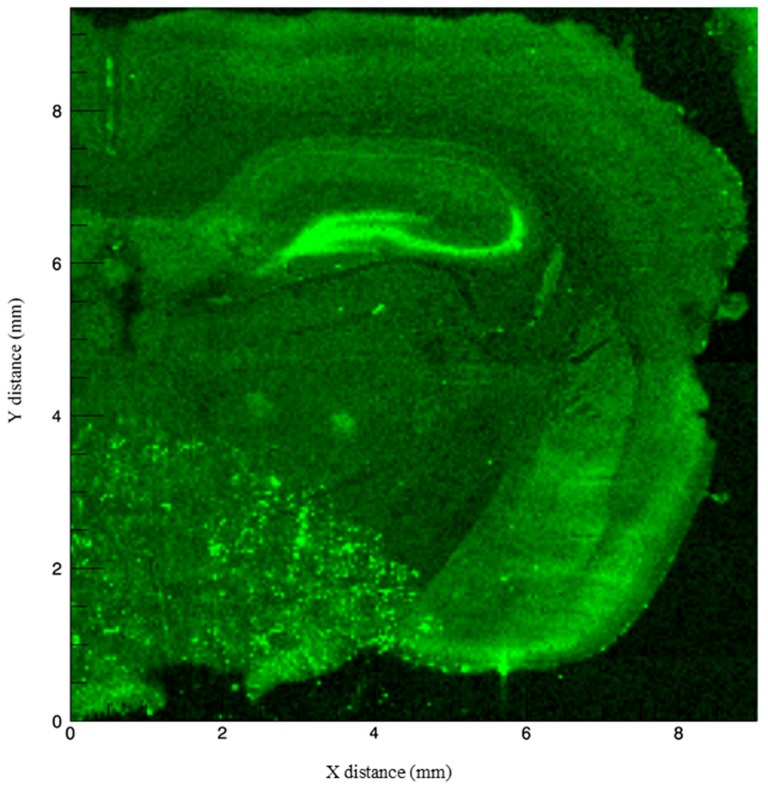
Fluorescence intensity of Zn normalized to the incident X-ray intensity. The strongest signal arises in the granule cell layer of the dentate gyrus and the CA3 pyramidal cell layer. Notable signal is also seen in the piriform cortex (lower right cortical regions of the hemisphere).

**Figure 5 brainsci-09-00074-f005:**
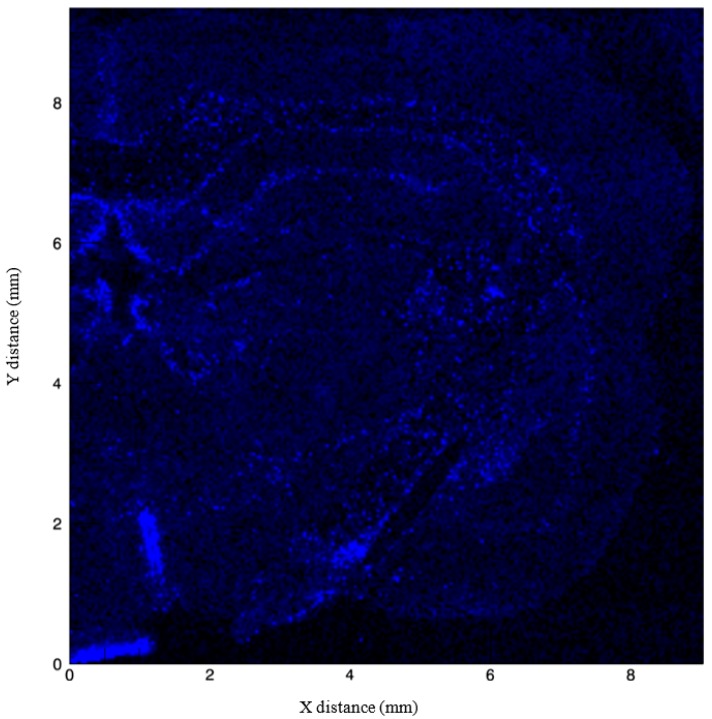
Fluorescence intensity of Cu normalized to the incident X-ray intensity. Copper is localized to areas surrounding the corpus callosum, the linings of the third ventricle, and the choroid plexus.

**Figure 6 brainsci-09-00074-f006:**
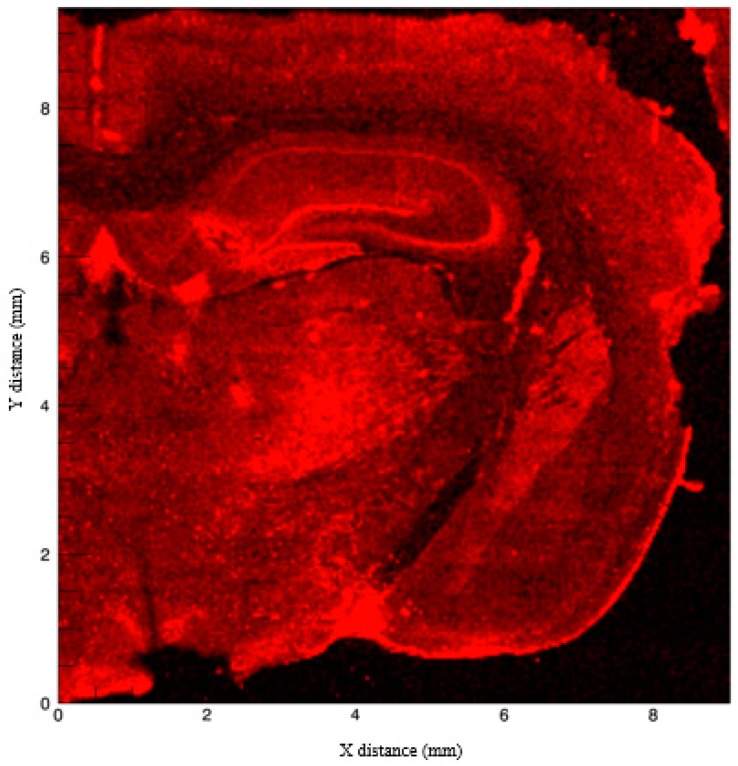
Fluorescence intensity of Fe normalized to the incident X-ray intensity. Strong signals arise from the basal ganglia where generally higher levels are reported. The CA3 region and dentate gyrus also show notable fluorescence.

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
