# Peer review of "Localization of Free and Bound Metal Species through X-Ray Synchrotron Fluorescence Microscopy in the Rodent Brain and Their Relation to Behavior"

_brainsci, 2019, doi:10.3390/brainsci9040074_

Round 1
Reviewer 1 Report
line 156 fused quartz slides were also Cu and Fe free, yes?
line 471 can lead "to" disruptions
Under acknowledgements, add " Use of the National Synchrotron Light source, Brookhaven National Laboratory, was supported by the U. S. Depart. of Energy, Office of Science, Office of Basic energy Sciences, under Contract No. DE-AC02-98CH10886. I don't know if tony and Steve want something added for X26. I'll leave that up to them
Author Response
We thank the reviewer for his/her comments on our submission. Per the reviewer's suggestions, we have made the following revisions that are highlighted (in yellow) in the updated manuscript:
Line 164 (re: slides): we confirmed with contributing author A.L. that the brain samples were mounted on to "metal-free, Suprasil quartz slides". This has been updated in the manuscript.
Line 471 (error): "to" has been added to this sentence.
Acknowledgements: the acknowledgements section has been updated to include supporting contract information.
Additional minor spelling errors, references, and other comments from the second reviewer have been addressed in the updated manuscript.
Thank you.
Reviewer 2 Report
This review manuscript entitled “Localization of free and bound metal species through X-ray synchrotron fluorescence microscopy in the rodent brain and its relation to behavior” presents a comprehensive literature covering the presence and role of essential bio-metals in brain areas with particular emphasis on hippocampus, a region critically involved in learning and memory. The authors convincingly suggest further research to fill the gaps in our understanding of bio-metals in brain physiology and pathophysiology. I appreciate authors effort to diligently introduce several disparate results opposing different views with possible reasoning, wherever required in the manuscript. Overall, the present review is very well written, easy to understand and provides all-inclusive information about 3 most abundant bio-metals in brain. However, to support some statements and to shed light on some contentious debates, some additional references and literature review is necessary. I have following comments that I feel will further improve this review article:
1. Please add references for this very first sentence of the review article: “Metal homeostasis has gained attention within the past few decades due its involvement in numerous biochemical interactions and diseases, such as Wilson’s disease, Menke’s disease, Alzheimer’s disease, depression, and anemia.”
2. Line 209: “further detailed Zns role in neurotransmission notable in forebrain structures including”. Replace with “further detailed role of zinc in neurotransmission, notable in forebrain structures including”
3. Fe, Zn, Cu and Mg share several properties like being divalent cations, occasionally using the same transport systems and compete to pass through same ionotropic channels. Common ion effect should be considered while interpretation of experiments that used externally administered metal salts. Please discuss this after line 517.
4. Figure 5: I have hard time to follow the legends and the image. I could not localize CA1 and CA2 mentioned in the legend. I suggest either keeping a cartoon diagram or the image presented in Fig 3 in at least Fig. 5.
5. Please check that all references were added at the end of the article. I could not find reference no. 137.
6. Line 471: “development, can lead disruptions in morphological development, axonal myelination, and”. Replace it with “development, can lead to disruptions in morphological development, axonal myelination, and”.
7. Line 524: “and CA3 field, which is known to be a NMDA-independent region of the hippocampus”. I am not convinced by this statement. Please see: Tarek Rajji et al, 2006; Nakazawa K et al., 2003; Ming-ChingChiang et al., 2018, etc.
8. Line 533: The role of bio-metal in neurogenesis has been suggested in a couple of instances in this review but adding a few more references with some facts will help establish the indispensable role that trace bio-metals play. Please see : Cathy W. Levenson et al 2011; Sherleen Fu et al 2015, etc.
Author Response
We thank the reviewer for his/her review of our submitted manuscript. Per the reviewer's suggestions, we have made the following revisions (highlighted in yellow) in the updated manuscript:
Additional references (Ayton, 2013; Bertinato & L'Abbe, 2004; Levenson, 2006; and Gozzelino & Arosio, 2016) have been added to the first sentence to support this statement.
Line 217 (re: "...further review the role of Zn..."): this sentence has been revised per the reviewer's suggestion.
Line 547: Commentary and a few examples have been added to note shared transport mechanisms and interdependent effects on ionotropic receptors.
Figure 5: In order to avoid any confusion, we have elected to delete the phrase "... signal arises from the CA1 and CA2 regions" from the figure caption. Although readers will have the option to zoom in on the image (if using an electronic device), we would rather emphasize the Cu concentrations around the SVZ and corpus callosum, as the strongest signal arises from these regions. We believe that the body of text provides enough background information regarding Cu concentrations in the hippocampus.
White et al., 2016 (previously reference 137, now 149) was added to the manuscript.
Line 497 (re: "... development, can lead disruptions..."): this error has been fixed.
Line 562 ("...NMDA-independent region of the hippocampus."): This sentence has been updated to reflect that this portion of the trisynaptic circuit is generally considered to exhibit LTP that is NMDA-independent. In addition, we have added additional text and references to the main body of the text (Lines 133) to discuss alternative mechanisms of LTP in the mossy fibers which could, in fact, be mediated by NMDA receptors, and their roles in learning and memory using the references provided.
Each metal section has been revised to include brief commentary about metals' role in neurogenesis (Zn - line 312; Cu - line 382; Fe - line 502)
Minor changes in syntax, minor grammatical errors, and references were addressed/added to the manuscript. Additional comments were also included per Reviewer 1's suggestions. Thank you.